# Recruiting strategic human capital from MNCs—Does hiring MNC managers enable exporting in domestic firms?

**Pedro de Faria**[1☯], **Torben Schubert**[2,3☯]*, **Wolfgang Sofka**[2,4,5☯]

**1** Faculty of Economics and Business, University of Groningen, Groningen, The Netherlands, **2** Department of Design Sciences, CIRCLE - Centre for Innovation Research, Lund University, Lund, Sweden, **3** Fraunhofer Institute for Systems and Innovation Research ISI, Karlsruhe, Germany, **4** Department of Strategy and Innovation, Copenhagen Business School, Frederiksberg, Denmark, **5** Strategy, International Business and Entrepreneurship Group (SIBE), University of Liverpool Management School, Liverpool, United Kingdom

☯ These authors contributed equally to this work.
* torben.schubert@circle.lu.se

**Data Availability Statement:** Data cannot be shared publicly because of the Swedish Statistical Office is the owner of the data. Description of the data set and the third-party source: The data

## Abstract

Exporting is a central growth strategy for most firms and managers with international experience are instrumental for export decisions. We suggest that such managers can be hired from Multinational Corporations (MNCs). We integrate theory from strategic human capital research into models explaining export decisions. We theorize that hiring managers from MNCs increases the odds of domestic firms to start exporting and this effect depends on the similarities between hiring firms and MNCs. We hypothesize that young firms will benefit comparatively less from hiring MNC managers. In contrast, firms with internationally diverse workforces and with high degrees of hierarchical specialization will benefit the most from hiring MNC managers. We test and support these hypotheses for 474,926 domestic firms in Sweden, which we observe between 2007 and 2015.

## 1. Introduction

Exporting is a fundamental growth strategy for many firms [1–3]. One key to having competitive products and processes for export markets originates from domestic firms' endowment with internationally experienced managers [4–6]. If firms can increase their capacity of managers with international knowledge, skills, experiences and other characteristics, for short human capital [7], they also increase their chances for exporting. While the role of labor markets to achieve strategic goals has been emphasized in other fields of management [8–10], the link between internationally experienced managers and exporting has not received great theoretical attention in international business, with the exception of the very specific case of entrepreneurs [5, 11, 12].

In this study, we focus on a particular hiring strategy of domestic firms: targeting managers who work for firms in which international experiences are typical, i.e. Multinational Corporations (MNCs). More specifically, we investigate how export decisions of domestic firms are influenced by the hiring of former MNC managers. For this purpose, we integrate theoretical

structure underlying the publication is a huge interlinked national registry dataset covering information on Swedish firms and the Swedish population over time. It can be understood as a rich linked employer-employee database, which allows for panel coverage. The data underlying the publication is owned by the Swedish Statistical Office (SCB) and hosted in the MONA-system. The MONA system is a remote access data infrastructure, from which only aggregated results (tables, regressions, graphs etc.) but no unprocessed raw data can be extracted. Publicly sharing a minimal dataset is therefore legally not possible. However, access to MONA is principally possible for research organizations. Obtaining access requires application because the information contained in MONA pertains also to sensitive individual data requiring personalized access to be able track and control data treatment. How to get access: SCB grants automatic full data access to employees of organizations, who already have access to MONA. If the employee's organization has no access to MONA, applications can be directed to the MONA-group in SCB (mona@scb.de). The application process entails costs and therefore may not be recommendable for the sole purpose of a replication. For replication purposes, we encourage requesting (temporary) membership at the corresponding author's home organization CIRCLE - Centre for Innovation Research. Affiliated members will the full rights to access the CIRCLE-data-infrastructure including the full raw data underlying this publication. Request for (temporary) affiliated membership should be addressed to Markus Grillitsch, markus. grillitsch@keg.lu.se and Torben Schubert, torben. schubert@circle.lu.se.

**Funding:** TS: VINNOVA, 2017-05382, www. vinnova.se, the sponsor played no role in the study design or the preparation of the manuscript.

**Competing interests:** The authors have no competing interests to declare, which are relevant to this study. Torben Schubert received funding from VINNOVA for the purpose of this project within a research grant. This does not alter our adherence to PLOS ONE policies on sharing data and materials.

mechanisms from strategic human capital literature [8, 13, 14] into models explaining export decisions of firms based on the international experience of their management [5, 15, 16]. We establish the reasoning for how the average domestic firm hiring managers from MNCs increases its likelihood to start exporting. Extant literature acknowledges the challenges and uncertainties associated with exporting. Domestic firms need to close knowledge gaps about demand, technology and resources requirements as well as competition on foreign markets [17]. Hence, firms prepare for exporting by improving technologies [18, 19] for better process and product quality [20, 21] including the development of product innovations [22–25]. Logically, such decisions require informed management. Subsequently, we take into account that firms are heterogeneous in the degree to which they can benefit from human capital that was created in other firms [26]. In specific, we theorize that the strength of the effect on the likelihood to start exporting depends on whether the hiring domestic firm shares organizational similarities with MNCs because of boundaries to effective human capital transfer across organization contexts [13]. We focus on dominant characteristics of MNCs and hypothesize that young firms have a lower likelihood to start exporting after hiring former MNC managers while international diversity and hierarchical specialization within domestic firms increase this likelihood from hiring former MNC managers.

We utilize a unique empirical opportunity provided by linked employer-employee data on the population of Swedish firms covering the period 2007–2015 collected by Sweden's statistical office Statistics Sweden (SCB) to test our hypotheses. Our final dataset covers 474,926 domestically-owned firms in Sweden. The dataset allows us to identify (a) the population of domestic firms that is not yet exporting, (b) their hiring of managers from MNCs (treatment), (c) a matched control group as well as (d) firms starting to export. We rely on entropy balancing to account for potential biases originating from the selectivity of the hiring decision [27, 28]. In line with our theoretical predictions, we find that domestic firms increase their likelihood to start exporting when they hire managers from MNCs and that this effect is weaker for domestic firms that are in their startup phase but stronger for firms with an internationally diverse workforce and a high degree of hierarchical specialization.

Our findings have two main implications for academic research. First, recent literature on export decisions emphasizes how exporting is not merely a function of a firm's resources and productivity but depends significantly on the human capital preparing and conducting export decisions [5, 29, 30]. We establish the hiring strategies of domestic firms as an important underpinning of export decisions.

Second, strategic human capital literature has increasingly moved away from describing the value of human capital for different organizations along the dimensions of strictly firm-specific or generally valuable for all firms [31]. Instead, some organizational contexts enable individuals to reveal and develop human capital that is scarce and valuable across firm and industry boundaries [32]. However, these transfers of human capital across firm and sector boundaries are unlikely to be seamless [13]. Our theoretical model incorporates both the benefits of hiring from MNCs for accumulating international human capital as well as the barriers for utilizing it in domestic firms. Future studies can build on this theory and explore other specificities of MNCs as work places, which make it hard to hire from them.

## 2. Theory and hypotheses

Our theoretical reasoning is at the firm level and predicts differences in the likelihood of domestic firms to start exporting. We develop a theoretical logic for how hiring former MNC managers increases the probability of a domestic firm to start exporting and identify boundary conditions for this effect based on defining characteristics of MNCs.

## 2.1 The nature of export decisions

Internationalization strategies of firms typically start with the commercialization of products and services in foreign markets, i.e. exporting [1, 33]. Exporting entails comparatively more flexibility and fewer risks when compared with resource-intensive internationalization strategies like Foreign Direct Investment (FDI). Still, the transition from a domestic strategy to one involving foreign markets is more complex than a simple increase in production capacity [4] and requires considerable investments [34]. In order to be competitive on foreign markets, firms need to invest into better products and more efficient processes since they will face a broader set of competitors [35, 36]. Furthermore, when entering export markets, firms often need to adapt their products to local needs and their internal organization to foreign clients [37]. That is, starting to export is a challenging and consequential strategic decision, which does not only affect a firm's sales department [4, 25].

Existing literature has stressed the role of human resources in the decision-making and implementation of exporting strategies [4]. As highlighted by [5], export orientation and export performance are not only dependent on the age, size and resources of a firm but also on the characteristics and past experiences of its managers. Internationalization decisions of managers build on their knowledge on how to prepare the organization to enter foreign markets [11]. Firms benefit from having managers with the experience in evaluating business opportunities and strategic decision making [38].

Firms with experienced managers are likely to make better internationalization decisions. Decisions to internationalize are inherently uncertain in nature since customer tastes, demands, regulations and competing products abroad are difficult to predict ex ante [17, 39]. The decision to start exporting is dependent on the management's ability to recognize opportunities for expansion to foreign markets and prepare the firm for internationalization [5]. Firms with internationally experienced managers are better able to develop expectations and build assumptions about the potential problems and opportunities associated to foreign markets [12, 40, 41]. Such human capital is developed over time and shaped by the background as well as experience of managers in activities with an international dimension [40, 42]. Work experience with an MNC can constitute such international experience.

## 2.2 Hiring from MNCs and its contribution to export decisions

Human capital consists of knowledge, skills, and abilities possessed by individuals and that can be useful for firms [7, 43]. The creation of human capital occurs within an organizational context that allows individuals to make use and develop valuable human capital [32]. Working for a firm exposes employees to unique learning opportunities from firm technologies [44], routines [45], capabilities [46], teams [47] and networks of clients [48] or stakeholders [49]. While certain parts of this human capital remain specific to the organizational context in which it was created, many aspects are valuable to other firms hiring employees with particular work experiences [32]. Additional evidence for this claim is provided by [50].

Positive performance outcomes for hiring firms are likely to emerge when newly hired employees have built up knowledge and skills for particular tasks that they can fruitfully apply in the new firm [13]. Hiring human capital across organizations has been shown to benefit firms' knowledge search [51–53] and innovation outcomes [54], the performance of startups [55], favorable implementation of regulations and standards [56] as well as social networks [57]. We reason that MNCs provide an organizational context in which employees develop human capital that can facilitate export decisions when they are hired by domestic firms.

During their tenure at an MNC, employees accumulate specific knowledge and experience that individuals with exclusively domestic careers can hardly obtain [58]. Gaining experiences

navigating in multicultural and international work environments allows individuals to develop a particular type of human capital, described in the literature as international human capital [59]. International human capital is associated with the ability of employees and managers to learn from and adapt to diverse business and cultural practices [60]. While working for MNC subsidiaries, employees leverage their experience and knowledge in an international environment and broaden the way in which they search for business or technological solutions [61]. Moreover, managers working in multinational environments develop particular skills, work routines and tacit knowledge, like managing international teams or the ability to accommodate cultural differences [62–64]. Obviously, not all MNC managers accumulate such international human capital to the same degree and some jobs or projects are more international than others. It is important for the purpose of our reasoning that (a) MNCs provide opportunities for creating international human capital and (b) hiring firms can observe the international dimension of MNC work experience when they hire. If other firms would create similar opportunities for creating international human capital, we should not find any significant effects for hiring from MNCs in our empirical study.

The international experience of managers [16] and founders [12] is an important driver of the export activities of firms. Hiring managers who have previously worked for an MNC allows domestic firms to acquire international human capital that is particularly relevant for their internationalization [5, 65]. International experience provides managers with opportunities to obtain knowledge about foreign markets and on how customer needs differ across countries [66] [67]. It also allows managers to develop skills on how to evaluate business opportunities abroad and screen promising export markets as well as to gain experience working together with foreign partners [15]. Moreover, as noted by [11] and [65], working in international environmenagers' ability to develop business practices that may help domestic firms to start exporting. In sum, we conclude that hiring MNC managers increases the capacity of a domestic firm for recognizing and assessing opportunities on foreign markets. As a result, these domestic firms become increasingly likely to start exporting. We propose:

**Hypothesis 1 (H1)**. Domestic firms hiring MNC managers have a higher likelihood to start exporting than comparable domestic firms do.

## 2.3 Organizational contexts and the strength of export effects of hiring former MNC managers

Our baseline expectation predicts that the average domestic firm increases its probability to start exporting when it hires MNC managers. Somaya et al. [48] suggest that the organizational context of the hiring firm is an important factor influencing the relationship between employee mobility and firm outcomes. Accordingly, we explore how hiring MNC managers into different organizational context affects the likelihood to start exporting. Theoretically, we ground our reasoning in the differences in organizational contexts in which the international human capital of MNC managers was created and in the domestic firms in which it is applied. Dokko et al. [13] discuss in this regard how work experience in an organization does not just create valuable human capital but also shapes the convictions and behaviors of employees which can in turn create detrimental frictions when applied outside of the original context in hiring firms.

MNCs are likely to generate human capital in their employees that remains specific to the MNC context and is hard to transfer to domestic firms. These particularities have often times been identified as the dual work context of MNC managers combining intra-MNC with

country level demands [68] or persistent "liabilities of foreignness" that keep MNCs from completely adopting processes and procedures of their host countries [69]. If work contexts appear superficially similar but have important structural differences, frictions are likely to emerge [70]. Under such conditions, the potentially valuable human capital acquired in one firm will not reach its full performance potential in another firm. Instead, hiring firms are more likely to experience frequent errors, delays and conflicts. We will reason that such "negative transfers" [13] of otherwise valuable human capital from MNCs to hiring domestic firms are more likely when hiring firms differ structurally from MNCs.

In the following, we explore three characteristic dimensions of heterogeneity between MNCs and domestic firms which are likely to affect the export effects of hiring MNC managers, i.e. the maturity of the hiring firm, the international diversity of its workforce and its degree of hierarchical specialization. While these characteristics are essential for human capital development in MNCs, we do not claim that this is an exhaustive list of moderating factors. We choose them to establish the overarching insight that the export effects of hiring MNC managers are affected by the organizational context of the hiring firm based on empirically testable hypotheses.

## 2.4 Young, domestic firms and the export effects of hiring from MNCs

We start out by considering the situation of young, domestic firms hiring MNC managers. Firms go through organizational lifecycles [71]. Young firms find themselves in a startup phase which is characterized by uncertainty and dynamic changes of products and processes [72]. During this stage, firms typically lack formal procedures and rely heavily on their founders. Founders are usually the dominant decision makers in startups and the level of ownership legitimizes their decision making [73]. In this sense, founders are typically the major sources for innovation and creativity in their startups [74]. Task specialization is rare in startups and most managers deal with broad sets of responsibilities [75]. Typically, the startup stage of young firms ends when they have reached a certain threshold of increasingly stable and predictable business conditions which allows them to professionalize decision making [72]. While being organizationally young is certainly a qualitative assignment differing between firms, there are some indications that the early-startup phases may last between four to seven years after founding [76]. Our definition of domestic firms is that they are neither owners of foreign subsidiaries nor themselves owned by a foreign parent company.

We reason that hiring MNC managers is comparatively more likely to help domestic firms starting to export once they are older and have crossed the threshold from startup firms for two reasons. First, substantial parts of the skills that managers have acquired while working for MNCs will not be valuable in the startup context. MNCs can rely on resources and technologies from other subsidiaries [77, 78]. Hence, MNC managers secure access to these resources by complying with intra-MNC standards [68]. In contrast, startups benefit from dynamic decision making and rapid changes given uncertain business conditions [72]. They are typically resource constrained and need to find ways to extract maximum value from the resources at hand [79]. They do that often by relating to other firms outside of their own organizational boundaries [80]. Unlike resource exchanges inside MNC structures, exchange interfaces for young firms are more instable because of a lack of intra-corporate control structures. More generally, young firms almost inevitably face so-called liabilities of newness pertaining to lack of tangible and intangible resources as well as lower levels of institutional legitimacy or trust [81, 82]. Managing these liabilities of newness is often complex and requires capabilities, which MNC managers may not possess.

Second, most startups lack task specialization. Startup decision makers are typically required to pay attention to a whole range of activities ranging from managing current operations to innovating new products [83, 84]. Only as the start-ups grows, it moves to increasing internal division of labor with a need to professionalize [85], e.g. by relegating managerial tasks from the founding team to professional managers [76, 86]. In contrast, MNC managers are more likely to apply skills that are shaped by specialization in which certain subsidiaries are responsible for specific tasks while relying on other subsidiaries or headquarters for others [87].

In sum, we conclude that the positive effect of hiring MNC managers by domestic firms for starting to export will be comparatively lower for young domestic firms. We propose:

**Hypothesis 2 (H2)**. Domestic firms hiring MNC managers have a higher likelihood to start exporting than comparable domestic firms do, and this effect is weaker for young domestic firms.

## 2.5 International workforce diversity in domestic firms and the export effects of hiring from MNCs

We focus on the international diversity of the workforce of domestic firms as a second factor influencing the effect of hiring MNC managers on the likelihood of a domestic firm to start exporting. MNCs are almost by definition internationally diverse organizations since they have subsidiaries located in various countries and their workforce often includes host country nationals, expatriates from the MNC's headquarters and individuals with international careers [88]. Individuals from different countries of origin have different sets of experiences that shape their mental models [89]. Therefore, international diversity is translated into heterogeneity in perspectives, values, cognitive schemas, and behaviors [90–92]. The cognitive conflicts associated with this heterogeneity have mixed effects on organizations and create unique environments. On the one hand, such conflicts stimulate creativity, learning, and consequently lead to improved innovation processes [93, 94]. On the other hand, diversity can lead to clashes between individuals and to nationality-based categorization that leads to affective conflicts [94, 95].

Domestic firms with an international diverse workforce provide a work environment that is more in line with the one from MNCs. Compared with homogenously staffed domestic firms, internationally diverse domestic firms will have a workforce with a higher level of heterogeneity in perspectives, values, cognitive schemas, and behaviors [90]. This creates a work environment that is likely to have the cognitive conflicts commonly present in MNCs. Internationally diverse domestic firms and MNCs share the opportunities associated with diversity, like creativity [93]. That is, when joining an international diverse domestic firm, former MNC managers will encounter an organizational setting that benefits from the skills acquired in the MNC subsidiary context [13]. Put differently, former MNC managers hired by domestic firms lacking such international diversity are likely to experience how their creative solutions originating from MNCs are inefficient or ineffective.

Consequently, we predict that domestic firms with internationally diverse workforces will be better able to take advantage of the ability of former MNC managers to recognize and assess opportunities in foreign markets. Thus, we propose:

**Hypothesis 3 (H3)**. Domestic firms hiring MNC managers have a higher likelihood to start exporting than comparable domestic firms do, and this effect is stronger for domestic firms with internationally diverse workforces.

## 2.6 Hierarchical decision-making in domestic firms and the export effects of hiring from MNCs

Finally, we focus on differences in the degree to which MNCs and domestic firms rely on hierarchical decision-making. Hierarchical differentiation in the decision-making of organizations reflects the need for coordination [96] and the organization of intra-firm collaboration including the joint use of resources. Because hierarchy relies on appropriately informed managers, the degree of hierarchical differentiation is intimately tied to the innate limitations of managers to gather, keep, process and transmit information. Organizations counter these limits in information processing capacities of individual managers by introducing middle management levels between top management and workers executing tasks [97]. Middle managers are typically specialized administrators to whom specific decision rights have been delegated [98]. Such specialization enables parallel information processing in organizations which in turn increases information processing capacities [99]. Besides, middle management frees up the attention of top management for strategic decisions while middle managers can handle routine decisions within their field of responsibility [100].

MNCs rely heavily on hierarchical decision-making. Most subsidiaries have dedicated mandates for the type of activity that they are supposed to perform within the MNC network [87]. Global headquarters may encourage initiative taking in some subsidiaries [101], but it typically also constrains their autonomy [102, 103]. While domestic firms may also rely on hierarchical differentiation, it is more consequential in MNCs in which top management is typically geographically remote. Taken together, we suspect that the skills that MNC managers have acquired while being exposed to the hierarchies within MNCs, will be less useful when they are being hired by domestic firms with flat hierarchies. Instead, we propose that former MNC managers will be more likely to help starting export activities when they apply their skills in hierarchical domestic firms. We hypothesize:

**Hypothesis 4 (H4)**. Domestic firms hiring MNC managers have a higher likelihood to start exporting than comparable domestic firms do, and this effect is stronger for domestic firms with a high degree of hierarchical specialization.

## 3. Empirical study

### 3.1 Data

We test our theoretical predictions using a linked employer-employee dataset on the population of Swedish firms for the period 2007–2015. The choice of the time period follows data availability. 2015 was the most recent year available within the infrastructure of the data. Sweden is a particularly fitting country setting for studying MNCs and the export decisions of domestic firms because it is a small open economy with high levels of integration into global value chains. This openness is reflected in the Eurostat figures on foreign direct investments, which amount to 64% of GDP in Sweden as compared for example to Germany with only 24%.

The data is collected on a yearly basis by Sweden's statistical office Statistics Sweden (SCB). SCB provides various types of firm and individual-level information in separate databases, which can be merged through the use of common firm and individual identifiers. In this study, we use six different databases: (1) the business statistics database (FEK), (2) the business group register database that provides basic firm-level and group structure information; (3) the integrated longitudinal database for health insurance and labor market studies (LISA) that

includes detailed individual level information on firms' employees including their mobility patterns; (4) the export in goods database; (5) the database on foreign-owned firms (Utland-sägda Företag); and (6) the database on Swedish-owned firms with subsidiaries in foreign countries (SVIK). The combined linked employer-employee database is a unique opportunity to study MNC employees' mobility and the export decisions of domestic firms. The quality of the data is evidenced by recent publications in leading management journals [104–106].

Our primary goal is to explain how the hiring of managers with MNC experience affects the likelihood of domestic firms to start exporting. Thus, we restrict the sample to fully domestically owned firms. Moreover, we further limit the sample to firms without exporting activities in the preceding three years of observation. This allows us to eliminate intermittent exporters from the sample which may otherwise bias the results [107]. Our databases provide annual firm-year (panel) observations covering the period from 2007 until 2015. The 3-year-non-exporting condition implies that our estimation sample effectively covers the period from 2010 to 2015. In total, we obtain data for 474,926 firms observed over the sample period. 33,294 of those firms have hired a total of 164,621 managers with MNC background and 11,016 domestic firms started exporting.

## 3.2 Variables

**3.2.1 Dependent variable.** Our dependent variable is the export status of the firm in any specific year. It is equal to one if the firm started to export goods in that year. Given that all firms in our sample have not exported in the three years prior, we will estimate the probability that their export status changes to one.

**3.2.2 Independent and moderating variables.** We are primarily interested in how hiring managers from MNCs affects the export status of domestic firms (Hypothesis 1). This requires the definition of MNCs. In our context, MNCs are either firms that are foreign majority-owned, i.e. the firm is the subsidiary of a foreign MNC in Sweden, or have a majority-owned Swedish ownership and subsidiaries abroad, i.e. are domestic MNCs. This broad definition of MNCs (domestic and foreign) constitutes a conservative setting for testing the hypotheses since domestic MNCs are likely to share similarities with domestic firms, i.e. the reference group. Hence, the ex-ante odds of finding significant results are lower. We conduct sensitivity checks which take the distinction between foreign and domestic MNCs into account and find consistent results for both types of MNCs. Further, we use the ISCO-88/SSYK-classification to identify MNC employees with managerial responsibilities: classes 1.* and 2.* that correspond to managerial and professional positions. Following these definitions, our independent variable is a binary variable that is one if a firm hired any individual that had managerial or professional positions in an MNC and zero otherwise. The definition of a dummy variable is useful since the vast majority of domestic firms in our sample does not hire former MNC managers during the observation period and very few hire more than one at the same time.

Testing hypotheses 2–4 requires the definition of three moderator variables. Our first moderator concerns the maturity of the hiring firm. The startup phase of a firm typically lasts about five to seven years after founding [105]. We reason that during this period firms have routines and procedures that are very dissimilar from the ones that can be found in MNCs. Thus, we create a binary variable that equals one if the firm is six years old or younger. For testing hypothesis 2, we interact this variable with our independent variable. We test for shorter (five years) as well as longer (seven years) time spans for defining startups and find consistent results, which are available upon request.

The second moderator is the international diversity of the hiring firm's workforce. Our data groups employee's according to their nationality: Swedish, Nordic countries, EU15,

Europe (except EU15 and Nordic countries), Africa, North-America, South-America, Asia, and Oceania. Based on this classification, we use the Teachman Entropy Index [108] as our diversity measure. The Teachman Entropy Index is comparable to the well-known Blau Index of diversity. It imposes a functional form that implies decreasing marginal returns to novelty, in the sense that adding novel non-existing backgrounds consecutively increases the diversity measure at a decreasing speed. The Teachman Entropy Index ranges from a minimum of zero to a maximum of -ln(K), where K = 9 refers to the number of categories. We interact the Teachman Entropy Index with our dependent variable to test hypothesis 3.

Our final moderator is the hiring firm's degree of hierarchical specialization. We measure it by considering the occupational differentiation within the firm. In particular, we take advantage of the fact that the ISCO-88/SSYK-classification orders occupations in decreasing order of complexity, skills, and responsibilities. In total, ISCO-88/SSYK includes nine hierarchically organized occupation levels. We define our measure of firm-level hierarchical differentiation as the number of hierarchical levels present in a firm. That implies that our variable can range from one, when a firm only has one hierarchical level, to nine, when a firm has all hierarchical levels. We interact this variable with our dependent variable to test hypothesis 4.

**3.2.3 Control variables.** We include a number of control variables, which existing literature has indicated that influence export decisions. First, we control for firm size by including the number of employees (log). Firm size is typically correlated with export and internationalization activities in general [5]. We introduce three control variables that separate the general hiring effects from the hiring of MNC managers. First, we use the hiring ratio (turnover of employees as share of total employment) to account for the general hiring of the domestic firm. Second, we include a dummy variable for whether the firm has recruited from exporting domestic firms as an alternative mechanism by which domestic firms can obtain international human capital. Third, we include the share of newly hired managers from other domestic firms to account for the fact that the positive effect of hiring managers from MNCs might be due to their managerial skills rather than their international experience.

We include a variable measuring the share of employees with tertiary education to account for the knowledge intensity of the hiring firm, which might also affect the export decision [109]. In addition, we use the share of managers and professionals in the firm in order to account for the managerial overhead of the firm.

Moreover, literature points towards strong self-selection effects of high quality firms into foreign markets [110]. Hence, we include a number of firm specific characteristics, which could through various mechanisms affect a firm's export decisions. In particular, we include variables measuring capital intensity (capital stock per value added), investment share (capital investment per capital stock) and labor productivity (value added per employee).

Lastly, we include year dummies, sector dummies (based on NACE 1-digits), and education field dummies [106] in order to account for remaining unobserved heterogeneity not explicitly controlled for by the previously described variables.

## 3.3 Estimation strategy

The main purpose of our estimation strategy is to identify the effects of hiring managers from MNCs on the probability of domestic firms starting to export. A potential source of bias could originate from the hiring decision, i.e. it is unlikely to be exogenously determined. Some firms might be more attractive employers for former MNC managers or hire them purposefully. We address potential endogeneity in hiring decisions in two ways. First, we restrict the sample to domestic firms that were not exporting in the three pre-sample years (2007–2009). By

implementing this restriction, we make the sample more homogeneous and closer to be a controlled trial setting because the sample becomes considerably more homogeneous.

Second, and in order to increase the homogeneity of the sample even further, we employ an entropy balancing procedure. Entropy balancing provides a weighting strategy for making regression samples balanced with respect to hiring and non-hiring firms and therefore contributes to reducing the risk of endogeneity biases from unaccounted heterogeneity. Entropy balancing relies on the intuitive idea that weights for observations can be optimized so that the group of treated observations (firms having hired MNC managers) and the group of control observations are no longer significantly different. Entropy balancing approaches are similar to other matching approaches (e.g. coarsened exact or propensity score matchings) but do not require iterative decisions by the researcher to achieve balance. Since our dataset is exceptionally rich, we are able to balance for the complete set of employee, sector and firm-level characteristics discussed previously. Additionally, we include the average relative salary level for all employees compared to the industry average to control for the firm's ability to make competitive wage offers to former MNC managers. Table A1 in S1 Appendix provides the results of the balancing procedure showing the mean in the matching variables before and after balancing. The results show that balancing provides a control group that is no longer significantly different along the matching variables. Subsequently, we use the entropy balancing weights in our regressions.

Following these considerations for potentially endogenous hiring decisions, we estimate the effects on the export decision of hiring firms. We rely on a number of linear and non-linear estimation approaches in order to show the robustness of our findings. In general, the limited dependent variable character of our main dependent variable, export status, suggests the use of binary-response models like probit. We use such models mainly as a way to assess the robustness of the baseline regression. The reason is that our main interest is the estimation of interaction effects. Interaction terms in non-linear models are very difficult to interpret because the sign of coefficients of the interaction terms is not indicative of the direction of the moderation. For example, a positive interaction coefficient does, unlike in linear models, not usually imply that the moderation effect is positive, too [111]. While it is possible to obtain some indications on the direction of the interaction in binary response models by plotting the marginal effects at different moderation values, it is not possible to get reliable information on the statistical significance of the overall moderation effect. We therefore opt for linear interaction models, which allow for clear-cut statistical inference of the moderation effects.

## 4. Results

### 4.1 Main results

Table 1 provides descriptive statistics that allow for the characterization of the average firm in our sample. It highlights a particular strength of our empirical setting, i.e. the ability to obtain the population of firms that are not exporting ex-ante instead of a potentially pre-selected sample of firms that are likely or pre-disposed to start exporting. The average domestic firm has around four employees, and 30% of them have a college degree. Moreover, only around 1% of firms in our sample start exporting during the observation period. The comparatively low numbers on firm size, education level and share of exporters are due to the fact that the domestic firms are typically much smaller, employ less qualified personnel and are less oriented towards international markets. These are important structural attributes of firms which make their inclusion as balancing and control variables salient. 32% of the firms in our sample are young firms (six years old or younger). International diversity is relatively low with an index

**Table 1. Descriptive statistics.**

| Variable | N | Mean | SD | Min | Max |
|---|---|---|---|---|---|
| Exporter (d) | 1923620 | 0.01 | 0.09 | 0.00 | 1.00 |
| Hiring of MNC manager (d) | 1923620 | 0.05 | 0.22 | 0.00 | 1.00 |
| Young firm (d) | 1923620 | 0.32 | 0.47 | 0.00 | 1.00 |
| International diversity | 1923620 | 0.07 | 0.25 | 0.00 | 1.00 |
| Hierarchical specialization | 1923620 | 1.04 | 1.05 | 0.00 | 9.00 |
| Domestic exporter background | 1923620 | 0.20 | 0.40 | 0.00 | 1.00 |
| Employees | 1923620 | 3.88 | 22.66 | 1.00 | 7707.00 |
| Hiring ratio | 1923620 | 0.10 | 3.39 | -612.00 | 1095.00 |
| Capital intensity (in Tho. SEK) | 1923620 | 154.00 | 12000.00 | 10600.00 | 556000.00 |
| Investment share turnover (in Tho. SEK) | 1923620 | 0.00 | 11.14 | 4190.00 | 13800.00 |
| Share managers and professionals | 1923620 | 0.14 | 0.31 | 0.00 | 1.00 |
| Share with non-MNC management experience | 1923620 | 0.19 | 0.35 | 0.00 | 1.00 |
| Share tertiary employees | 1923620 | 0.29 | 0.41 | 0.00 | 1.00 |
| Labor productivity (in Tho. SEK) | 1923620 | 432.00 | 769.00 | 142000.00 | 282000.00 |

value close to zero, 0.07. The firms in our sample have flat hierarchies with slightly more than one hierarchical level on average.

Table 2 shows the correlation coefficients for all independent variables included in our models. The correlation values do not raise concerns regarding potential multicollinearity problems. Furthermore, the average variation inflation factor (VIF) is 3.1 and below the critical thresholds of 5 and 10 [112]. Given the longitudinal nature of our data, we conduct tests for serial correlation, heteroscedasticity, and reverse causality. The results of these tests do not give indications of any serious endogeneity or autocorrelation issues.

Table 3 presents the results of the regression models. Model 1 shows the results from the panel-probit regressions. Model 2 shows the results of the same model estimated with OLS. Both models include entropy balancing weights. All models include sector, year, and education-level fixed effects. These are the baseline models that include the control variables and the

**Table 2. Correlation table.**

| | 1 | 2 | 3 | 4 | 5 | 6 | 7 | 8 | 9 | 10 | 11 | 12 | 13 | 14 |
|---|---|---|---|---|---|---|---|---|---|---|---|---|---|---|
| Exporter (d) (1) | 1.00 | | | | | | | | | | | | | |
| Hiring of MNC manager (d) (2) | 0.04 | 1.00 | | | | | | | | | | | | |
| Young firm (d) (3) | -0.03 | 0.09 | 1.00 | | | | | | | | | | | |
| International diversity (4) | 0.02 | 0.04 | 0.06 | 1.00 | | | | | | | | | | |
| Hierarchical specialization (5) | 0.08 | 0.21 | -0.04 | 0.18 | 1.00 | | | | | | | | | |
| Domestic exporter background 9d) (6) | 0.07 | 0.16 | 0.10 | 0.38 | 0.15 | 1.00 | | | | | | | | |
| Employees (7) | 0.03 | 0.16 | -0.02 | 0.27 | 0.11 | 0.17 | 1.00 | | | | | | | |
| Hiring ratio (8) | 0.01 | 0.08 | 0.03 | 0.12 | 0.07 | 0.09 | 0.29 | 1.00 | | | | | | |
| Capital intensity (9) | 0.00 | 0.00 | -0.03 | 0.01 | -0.01 | 0.00 | 0.00 | -0.01 | 1.00 | | | | | |
| Investment share turnover (10) | 0.00 | 0.00 | 0.00 | 0.00 | 0.00 | 0.00 | 0.00 | 0.00 | 0.00 | 1.00 | | | | |
| Share managers and professionals (11) | 0.01 | 0.18 | 0.05 | 0.18 | -0.03 | -0.03 | 0.00 | -0.01 | -0.03 | 0.00 | 1.00 | | | |
| Share with non-MNC management experience (12) | 0.01 | 0.12 | 0.02 | 0.03 | -0.05 | -0.05 | -0.02 | -0.02 | -0.03 | 0.00 | 0.68 | 1.00 | | |
| Share tertiary employees (13) | 0.00 | 0.16 | 0.09 | -0.01 | -0.02 | -0.03 | -0.01 | 0.00 | -0.03 | 0.00 | 0.37 | 0.42 | 1.00 | |
| Labor productivity (14) | 0.02 | 0.06 | 0.00 | 0.06 | 0.00 | 0.04 | 0.02 | -0.04 | 0.28 | 0.00 | 0.07 | 0.08 | 0.05 | 1.00 |

**Table 3. Baseline probit regressions (marginal effects for probit; dependent var: Dummy: Exporter).**

| | (1) | (2) |
|---|---|---|
| | **Probit using entropy balancing weights** | **Linear model using entropy balancing weights** |
| Hiring MNC manager | 0.0028 (0.000) | 0.0082 (0.000) |
| Young firm | -0.0069 (0.000) | -0.0083 (0.000) |
| International diversity | 0.0033 (0.000) | 0.0048 (0.002) |
| Hierarchical specialization | 0.0027 (0.000) | 0.0054 (0.000) |
| Dummy: Domestic exporter background | 0.0062 (0.000) | 0.0093 (0.000) |
| Log employees | 0.0017 (0.000) | 0.0019 (0.000) |
| Hiring ratio | -0.0001 (0.000) | -0.0001 (0.000) |
| Capital intensity | -0.0000 (0.220) | -0.0000 (0.149) |
| Share managers and professionals | 0.0009 (0.174) | 0.0007 (0.326) |
| Investment share turnover | 0.0000 (0.147) | 0.0000 (0.279) |
| Share with non-MNC management experience | 0.0064 (0.000) | 0.0062 (0.000) |
| Share tertiary employees | 0.0042 (0.000) | 0.0055 (0.000) |
| Lagged labor productivity | 0.0000 (0.027) | 0.0000 (0.240) |
| Constant | Yes | Yes |
| Year dummies | Yes | Yes |
| Sector dummies | Yes | Yes |
| Education field dummies | Yes | Yes |
| Observations | 1923620 | 1923620 |
| #Firms | 474926 | 474926 |
| *Pseudo-$R^2$/$R^2$* | 0.104 | 0.020 |
| Average VIF | | 3.13 |

*p*-values in parentheses

main independent variable and allow us to test hypothesis 1, predicting that hiring former managers from MNCs increases the probability of a domestic firm to start exporting.

The results for hypothesis 1 are consistent across the linear and non-linear models and show that domestic firms hiring MNC managers have a higher likelihood to start exporting when compared with other domestic firms. The two models however do differ in size. The probit model shows an increase of 0.2 percentage points (b = 0.0028, pval = 0.000), while the linear approach shows an estimated increase of 0.8 (b = 0.0082, pval = 0.000) percentage points.

The results also show some interesting patterns with respect to the control variables. First, we find evidence that hiring from domestic firms with export experience also increases the likelihood of a firm to start exporting. In addition, the overall share of managers and professionals employed by a firm have a consistently positive effect on its propensity to start exporting. The positive coefficients suggest that general managerial or professional experience have a human-capital-related effect on export strategies working independently of skills resulting from international human capital. In this sense, the control variable is useful for isolating the hypothesized effects.

In hypotheses 2 to 4, we explore moderation effects associated to heterogeneity among domestic firms. We reason that the degree to which a domestic firm benefits from hiring MNC managers depends on its maturity (hypothesis 2), international diversity of its workforce

**Table 4. Interaction OLS regressions for H1-H3 (dependent var: Dummy: Exporter).**

|  | (1) | (2) | (3) | (4) |
|---|---|---|---|---|
|  | Dummy: exporter | Dummy: exporter | Dummy: exporter | Dummy: exporter |
| (Hiring MNC manager)*(Young firm) | -0.0105 (0.000) |  |  | -0.0091 (0.000) |
| (Hiring MNC manager)*(International diversity) |  | 0.0119 (0.005) |  | 0.0076 (0.089) |
| (Hiring MNC manager)*(Hierarchical specialization) |  |  | 0.0024 (0.000) | 0.0015 (0.011) |
| Hiring MNC manager | 0.0134 (0.000) | 0.0073 (0.000) | 0.0036 (0.000) | 0.0093 (0.000) |
| Young firm | -0.0072 (0.000) | -0.0083 (0.000) | -0.0083 (0.000) | -0.0073 (0.000) |
| International diversity | 0.0048 (0.002) | 0.0039 (0.014) | 0.0050 (0.001) | 0.0043 (0.007) |
| Hierarchical specialization | 0.0053 (0.000) | 0.0054 (0.000) | 0.0051 (0.000) | 0.0051 (0.000) |
| Dummy: Domestic exporter background | 0.0091 (0.000) | 0.0092 (0.000) | 0.0088 (0.000) | 0.0088 (0.000) |
| Log employees | 0.0018 (0.000) | 0.0019 (0.000) | 0.0018 (0.000) | 0.0017 (0.000) |
| Hiring ratio | -0.0001 (0.000) | -0.0001 (0.000) | -0.0002 (0.000) | -0.0002 (0.000) |
| Capital intensity | -0.0000 (0.119) | -0.0000 (0.153) | -0.0000 (0.118) | -0.0000 (0.107) |
| Share managers and professionals | 0.0011 (0.148) | 0.0008 (0.296) | 0.0004 (0.552) | 0.0009 (0.237) |
| Investment share turnover | 0.0000 (0.287) | 0.0000 (0.278) | 0.0000 (0.264) | 0.0000 (0.273) |
| Share with non-MNC management experience | 0.0060 (0.000) | 0.0062 (0.000) | 0.0067 (0.000) | 0.0064 (0.000) |
| Share tertiary employees | 0.0055 (0.000) | 0.0055 (0.000) | 0.0056 (0.000) | 0.0055 (0.000) |
| Lagged labor productivity | 0.0000 (0.237) | 0.0000 (0.241) | 0.0000 (0.240) | 0.0000 (0.238) |
| Constant | -0.0113 (0.000) | -0.0106 (0.000) | -0.0100 (0.000) | -0.0104 (0.000) |
| Year dummies | Yes | Yes | Yes | Yes |
| Sector dummies | Yes | Yes | Yes | Yes |
| Education field dummies | Yes | Yes | Yes | Yes |
| Observations | 1923620 | 1923620 | 1923620 | 1923620 |
| #Firms | 474926 | 474926 | 474926 | 474926 |
| $R^2$ | 0.020 | 0.020 | 0.020 | 0.020 |

*p*-values in parentheses

(hypothesis 3) and degree of hierarchical specialization (hypothesis 4). We present the results that allow us to test these hypotheses in Table 4 using linear regression models to account for moderation effects as explained in the estimation strategy section.

The interaction effect for whether the firm is six years or younger is -0.0105 (pval = 0.000) and highly significant (Model 1). In addition, international diversity moderates positively the effect of recruiting employees with MNC-backgrounds (Model 2). Likewise, the interaction with the number of hierarchical levels is 0.0024 (pval = 0.000) (Model 3). Moreover, Model 4 shows that the stated interaction effects remain significant, albeit at somewhat lower levels, when included simultaneously.

The moderation effects are plotted in Fig 1, in which the results indicate that the effects differ substantially across the distribution of the moderators. For example, for firms aged six years or less, the effect is very close to zero but is equal to about 1.5% in established firms. Similar observations can be made for international diversity, where the expected export-effect is 0.9% in firms with the lowest degree of international diversity and approximately 3% in the highest category. For firms with very low number of hierarchical levels, the effect is about 0.7% but increases to about 2.4% in the highest category. It should be noted however that in all cases, the baseline effect of recruiting employees with MNC-backgrounds remains significant. Overall, we find support for all hypothesized relationships and effects of substantial magnitude.

(a)

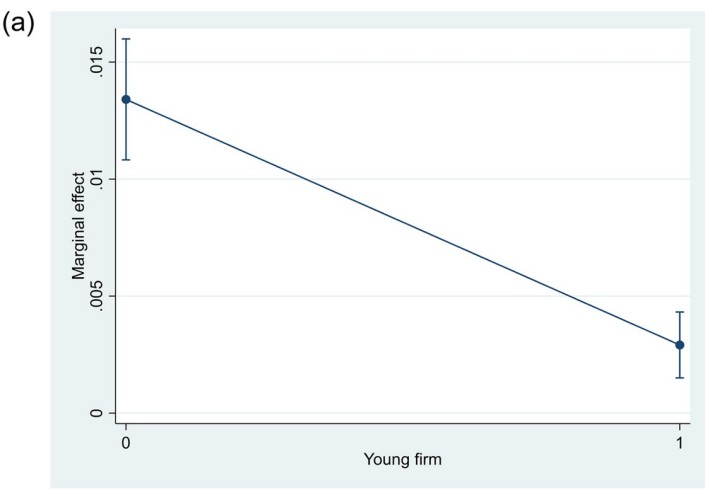

(b)

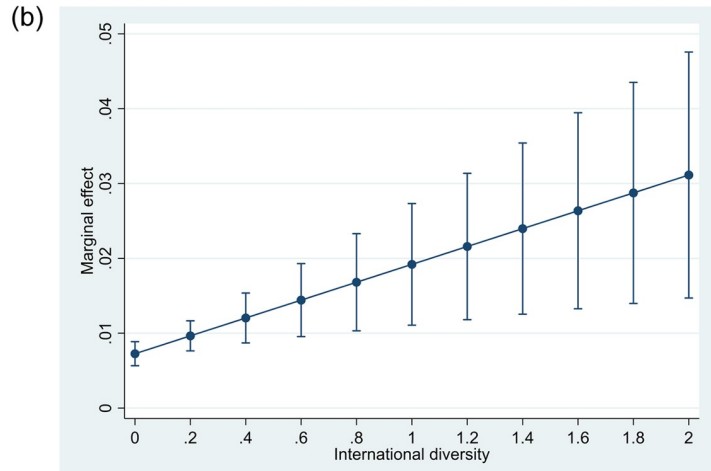

(c)

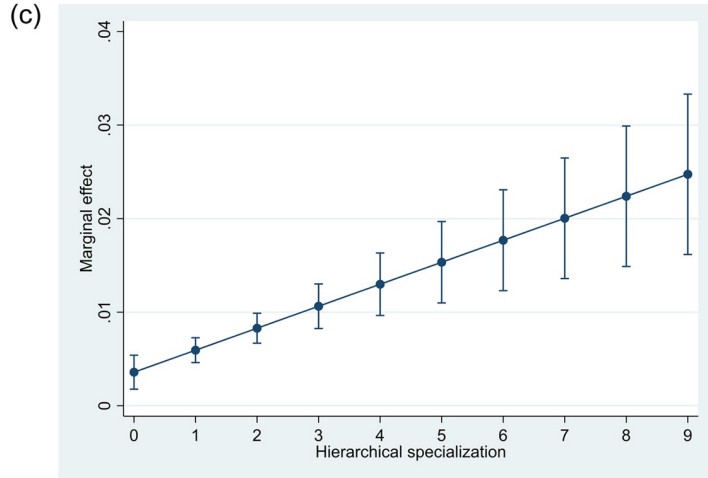

**Fig 1. a–c Visual representation of the moderation effects (with 95%-confidence intervals).**

## 4.2 Robustness checks

We conduct a number of additional estimations to demonstrate the robustness of the results. All results tables that are not explicitly referenced are available from the authors upon request. First, we replace the entropy balancing with an instrument variable approach, which can also take endogeneity biases from unobserved factors into account (see Table A2 in S1 Appendix). For obtaining instruments, we exploit the fact that some employees of MNCs had to look out for new employment due to the fact that the MNC subsidiary where they were working was closed down. Such displacements are useful for our purpose since they are not driven by negative performance of individual employees but by MNC-wide considerations. Displacement provides an exogenous shock on the labor supply and increases a domestic firm's likelihood to attract employees from MNCs. Therefore, it allows causal identification. We choose the entropy balancing approach for our main models since they allow more flexible testing of moderations, i.e. our hypothesized relationships. We note that we run only the baseline models as IV and probit models, because the interaction models are not easily compatible with these approaches. In particular, running an interaction model with IV would require us to instrument already two endogenous variables. In our case, the models become instable, which is usually the case even if instruments are strong. Moreover, the coefficients on the interaction terms do not have a simple interpretation in non-linear models. Most importantly, the t-test on the Null that the interaction coefficient is zero, is not a consistent test of the hypotheses that there are moderation effects. This conclusion holds only in linear models. The instrument variable regression support the hypothesis tests of the entropy balancing approach.

Second, we test for potential biases emerging from the definition of multinational firms. While the definition of foreign-ownership is quite unambiguous, the definition of a Swedish MNCs is less so since managers can both work in subsidiaries as well as global headquarters. Hence, we rerun all estimations and define MNCs more narrowly as foreign owned firms or Swedish headquarters of domestic MNCs. The results are robust, and in fact, become stronger rather than weaker.

Third, results may differ between foreign and Swedish MNCs. Thus, we repeat all estimations separating the two types of MNCs. While the baseline effects of recruiting from MNCs is somewhat higher for Swedish MNCs, the interaction effects support all hypothesized relationships for both types of MNCs.

Fourth, hypothesis 2 predicts a moderation effect from hiring firms in the startup phase. There is not commonly applicable firm age that corresponds with the length of a startup phase. We test hypothesis 2 in the main models by using six years as the threshold after which firms become increasingly mature. For consistency check estimations, we repeat all estimations defining the startup phase as either five or seven years of firm age. These additional estimations support hypothesis 2 as well.

Finally, another set of robustness checks concerns the sectors included in our sample. Export data includes trade in goods but not trade in services, so including services may be problematic. In a robustness check, we have excluded all service firms and find that the overall directions of the coefficients remain the same. Overall, the coefficients become somewhat larger in size, most likely because the risk of underestimating export activities is smaller in the restricted sample. We opt for keeping service firms in our main model estimations because firms classified in service sectors also export goods. Hence, testing our hypotheses for both manufacturing and service sectors is the conservative approach since it keeps firms in the sample with a lower likelihood to start exporting.

## 5. Discussion and conclusions

We conduct this study to shed light on the effects of hiring managers from MNCs for the export decisions of domestic firms. Existing literature has focused mostly on the hiring of MNCs from domestic firms [28, 113]. However, there is limited understanding of the theoretical mechanisms associated to the hiring of former MNC employees by domestic firms ([58, 114] are notable exceptions). We develop theory by integrating theoretical mechanisms from strategic human capital research [13, 32] into models export decisions of domestic firms [5, 15, 16].

We focus on MNCs as unique settings in which firms can find managers with international human capital [58] and reason that domestic firms hiring managers from MNCs will have a higher likelihood to start exporting. We theorize on how domestic hiring firms with organizational contexts similar to MNCs will be better able to take advantage of the international human capital of former MNC managers. We hypothesize that these effects will be weaker for younger and stronger for more international diverse domestic firms as well as for domestic firms with a higher degree of hierarchical specialization.

Our empirical findings support for our hypotheses. They have important consequences for academic research along two primary dimensions. First, the literature on export decisions has moved away from the idea that these decisions are mainly dependent on a firm's resources and productivity [5]. Instead, it highlights the importance of human capital and its international dimensions for how firms prepare to enter new markets [12, 15]. However, the source from which such human capital can be hired remain ambiguous. Our study advances literature on the human capital origins of export decisions by introducing their hiring decisions as important theoretical underpinnings. We demonstrate the usefulness of this theoretical angle for the hiring from MNCs but our theoretical model can inform future theorizing more generally on where firms can find international human capital and apply it productively, e.g. through global recruiting strategies.

Second, we introduce hiring from MNCs to domestic firms as a novel field for theorizing about the effects of MNCs on host countries. While a few studies have identified import flows of human capital from MNCs to domestic firms, they are typically investigating situations in which MNCs close down and require employee mobility [58, 114]. Our theoretical model is more general in nature since it identifies the specificity of human capital acquired in MNCs which can constrain its usefulness for domestic firms. We demonstrate the merits and boundary conditions for hiring from MNCs. On the one hand, MNC-specific human capital is valuable for domestic firms since it increases their ability to start exporting. On the other hand, organizational differences between MNCs and domestic firms constrain the positive effects. Overall, our theoretical reasoning can be usefully applied in future studies exploring alternative sources for utilizing international human capital, e.g. by hiring international consultancy firms, or strategic decisions of domestic firms such as offshoring activities abroad.

Apart from these implications for academic research, our findings inform the hiring decisions of domestic firms as well as the career decisions of MNC managers. For the former, we show that prospective exporters benefit from hiring MNC managers. However, our results also show that hiring firms need to have some structural features in place, i.e. a certain level of maturity, international diversity and hierarchical specialization, so that the newly hired managers can effectively guide the hiring firm towards exporting. For MNC managers, we show that their human capital has particular value for domestic firms which consider entering export markets. Besides, we offer a set of structural features of hiring firms that increase the odds for former MNC managers to succeed in turning domestic firms into exporters. They can adjust their career planning accordingly.

## 6. Limitations and future research

While conducting the study, we discover several fruitful pathways for future research that go beyond a single study. First, we develop a firm-level logic connecting hiring decisions with exporting. This perspective does not allow us to explore heterogeneity at the individual level. However, it is very likely that some employees of MNCs carry particular value for exporting in domestic firms, e.g. based on their nationality, export market experience, sales or marketing responsibilities. Similarly, some new hires might be particularly likely to suffer from conditions in hiring firms that do not allow them to put their international human capital to good use, e.g. a lack of international diversity. Hence, we encourage individual level studies that can explore the heterogeneity underlying the overall firm-level effects that we discover.

Second, we go to great length in isolating potential endogeneity biases originating from hiring decisions of domestic firms using both entropy balancing and instrument variable approaches. However, the hiring preferences of prospective exporters could constitute a valuable research question in itself. We separate hiring from MNCs from other sources of international human capital, e.g. other exporting firms, but future studies may focus on more fine-grained selection criteria, e.g. whether firms search managers to assess export markets, marketing experts to tailor products or engineers for advancing technologies.

Third, we identify structural features of hiring firms which limit the degree to which hiring MNC managers advances their export decisions. Our empirical results are consistent with the theorized moderations between the hiring of MNC managers and the organizational features of the hiring firm. However, the data material at hand does not allow us to identify the precise mechanisms underlying these moderation effects directly. Future studies might use dedicated research designs, such as experimental studies, to identify the mechanisms directly. Similarly, we focus on three prominent organizational features of MNCs that are comparatively less likely to exist in domestic firms. We do not claim that this is an exhaustive list. Instead, we encourage future studies theorizing other moderations effects and compare the magnitude of effects empirically, e.g. based on the top management characteristics of hiring firms.

Fourth, we benefit from a large dataset compiled for Sweden. Sweden is high-tech country, with many firms operating internationally and with flexible labor markets. Hence, it is suitable for testing hypotheses connecting hiring decisions with internationalization outcomes. Then again, future studies may not just produce comparative results for other countries but integrate mechanisms from labor market constraints or barriers to internationalization into our theoretical model. For example, the results for hiring firms in emerging markets might differ in the degree to which former MNC managers can transfer valuable human capital to domestic firms which are otherwise struggling to access such expertise.

Finally, our decision to construct a large and comprehensive dataset implied making trade-offs regarding time period covered. While we are confident that our structural theory describes aspects of organizing social life that are quite robust over time, we agree that testing our hypotheses with more recent data would be beneficial. Future studies may explore how the evolution of HRM practices (in particular associated to the Corona pandemic) may have affected the mechanisms that we explore in this study.

## Supporting information

**S1 Appendix.**
(DOCX)

**S1 File. Data sharing.**
(DOCX)

## Author Contributions

**Conceptualization:** Pedro de Faria, Torben Schubert, Wolfgang Sofka.

**Formal analysis:** Torben Schubert.

**Writing – original draft:** Pedro de Faria, Torben Schubert, Wolfgang Sofka.

**Writing – review & editing:** Pedro de Faria, Torben Schubert.

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
