## [Decision Letter · Decision Letter 0]

22 Jul 2021

PONE-D-21-19610

RECRUITING STRATEGIC HUMAN CAPITAL FROM MNCS – DOES HIRING MNC MANAGERS ENABLE EXPORTING IN DOMESTIC FIRMS?

PLOS ONE

Dear Dr. Schubert,

Thank you for submitting your manuscript to PLOS ONE. After careful consideration, we feel that it has merit but does not fully meet PLOS ONE’s publication criteria as it currently stands. Therefore, we invite you to submit a revised version of the manuscript that addresses the points raised during the review process.

We look forward to receiving your revised manuscript.

Kind regards,

Yuriy Bilan

Academic Editor

PLOS ONE

Journal Requirements:

[TS: VINNOVA, 2017-05382, www.vinnova.se, the sponsor played no role in the study design or the preparation of the maunscript]. 

We note that you received funding from a commercial source: VINNOVA

4. Please ensure that you refer to Figure 1 in your text as, if accepted, production will need this reference to link the reader to the figure.

Reviewers' comments:

Reviewer's Responses to Questions

**Comments to the Author**

1. Is the manuscript technically sound, and do the data support the conclusions?

Reviewer #1: Yes

Reviewer #2: Yes

2. Has the statistical analysis been performed appropriately and rigorously? 

Reviewer #1: Yes

Reviewer #2: Yes

3. Have the authors made all data underlying the findings in their manuscript fully available?

Reviewer #1: Yes

Reviewer #2: Yes

4. Is the manuscript presented in an intelligible fashion and written in standard English?

Reviewer #1: Yes

Reviewer #2: Yes

5. Review Comments to the Author

Reviewer #1: The choice of this time period requires justification (72-73)

It is necessary to dwell in more detail on the definition of "young, domestic firm" (222)

It is necessary to add other actions to work with information, "use" for example (291-293)

More comments see in the text.

Reviewer #2: The paper is scientifically sound and well-structured, main findings are justified using essential dataset and appropriate statistical tools of analysis.

However, the study has some drawbacks. The most essential from them is usage of the data of 2007-2015 as well as out-dated scientific literature to justify Authors' position and hypotheses to be checked. Combining not recent data and theoretical background for own research, Authors cannot prove that their results correspond to the current situation in HRM. If Authors could justify the unavailability of more recent data, this case can be appropriate to illustrate the methodology of links analysis in the field of HRM, particularly, hiring managers from MNCs, and their influence on economic activity of the firms, including export. In any case, the explanation of the 2007-2015 data use should be provided.

The Literature review in this thriving field of research should be updated without any doubts.

In the given form this research (despite its importance, scientific interest and carefully fulfilled analysis) looks like study been prepared earlier and not published.

More comments see in the text.

6. PLOS authors have the option to publish the peer review history of their article (what does this mean?). If published, this will include your full peer review and any attached files.

Reviewer #1: No

Reviewer #2: No

---

## [Author Response · Author response to Decision Letter 0]

30 Aug 2021

Dear reviewers,

thank you very much for your insightful comments, which have considerably improved the paper. We have dealt with them thoroughly and hope that we were able to deal with them satisfactorily.

We reproduce your detailed comments below and provide our responses in italics directly after each comment.

With best regards,

The authors

Reviewer #1: The choice of this time period requires justification (72-73)

We currently have access only to the years until 2015. Data before 2007 could not be used because the internationalization data is only available from this year onwards. We have explained these data limitations in the text in Section 3.1.

It is necessary to dwell in more detail on the definition of "young, domestic firm" (222)

We extended the literature work here following your suggestions. Following Extant research on life-cycle patterns of firms is not unambiguous about when firms are no longer considered as “young”, but prior studies have established that between 4-7 years is typically what would be considered as young. Domestic, for our purpose, are all firms, which either do not have foreign subsidiaries or are not owned by a foreign firm. This conservative definition of “domestic” is a useful choice for our research question which specifically investigates labour mobility from MNCs and benefits from an unambiguous definition of hiring firms which are not MNCs. These are also aligned with the definitions we adopted later on in our empirical analysis. Thus, we believe that the definitions of “young” and “domestic” are theoretically justified, firmly grounded in the literature and consistent with the empirical study. Following your guidance, we have expanded and clarified these considerations in the revised version of the manuscript.

It is necessary to add other actions to work with information, "use" for example (291-293)

Thank you very much for this comment. We realized that we were a little narrow here. We have discussed other types of functions or management tasks that are performed through hierarchical differentiation including the use of joint resources, collaboration and coordination in general. 

It [The reference list] should be updated in accordance with modern scientific research, some literature sources are outdated (1948, 1962, 1988, 1992…)

We followed your advice and updated the literature list with many very recent works. Indeed, some papers were quite old. Yet, we decided against dropping them, because several of them are quite fundamental and lay the foundations of research done today. We hope that this was in your interest. 

Reviewer #2: The paper is scientifically sound and well-structured, main findings are justified using essential dataset and appropriate statistical tools of analysis.

However, the study has some drawbacks. The most essential from them is usage of the data of 2007-2015 as well as out-dated scientific literature to justify Authors' position and hypotheses to be checked. Combining not recent data and theoretical background for own research, Authors cannot prove that their results correspond to the current situation in HRM. If Authors could justify the unavailability of more recent data, this case can be appropriate to illustrate the methodology of links analysis in the field of HRM, particularly, hiring managers from MNCs, and their influence on economic activity of the firms, including export. In any case, the explanation of the 2007-2015 data use should be provided. The Literature review in this thriving field of research should be updated without any doubts.

Thank you for this comment. Indeed, the period until 2015 is chosen because of limitations to data availability. No more recent data exists in the authors databases as of now. This is explained in Section 3.1. We also made an effort and explain this limitation explicitly as a well as the conceptual problems to infer to more recent periods in the limitations section. We agree that changing HRM practices can affect our results and therefore it is not a priori clear how far into the future our results will extend. At the same time, our structural theory describes aspects of organizing social life that are quite robust over time. To strengthen this claim, we followed your suggestion and updated our literature work substantially. We hope that this dual approach accommodates your concerns. 

In the given form this research (despite its importance, scientific interest and carefully fulfilled analysis) looks like study been prepared earlier and not published.

The research was conducted between 2019 and 2020. We presented earlier versions of the study in relevant seminars and conferences. Afterwards, we improved and revised the manuscript iteratively. Naturally, this process takes time but has hopefully elevated the quality of the manuscript that we could ultimately submit to the journal.

More comments see in the text.

We have incorporated all of your additional comments and literature recommendations in the text when they were immediately instructive and did not require a detailed reply.

---

## [Editor Report · Decision Letter 1]

14 Sep 2021

RECRUITING STRATEGIC HUMAN CAPITAL FROM MNCS – DOES HIRING MNC MANAGERS ENABLE EXPORTING IN DOMESTIC FIRMS?

PONE-D-21-19610R1

Dear Author

We’re pleased to inform you that your manuscript has been judged scientifically suitable for publication and will be formally accepted for publication once it meets all outstanding technical requirements.

Kind regards,

Yuriy Bilan

Academic Editor

PLOS ONE
---

## [Editor Report · Acceptance letter]

17 Sep 2021

PONE-D-21-19610R1 

Recruiting Strategic Human Capital from MNCs – Does Hiring MNC Managers Enable Exporting in Domestic Firms? 

Dear Dr. Schubert:

I'm pleased to inform you that your manuscript has been deemed suitable for publication in PLOS ONE. Congratulations! Your manuscript is now with our production department. 

Kind regards, 

on behalf of

Professor Yuriy Bilan 

Academic Editor

PLOS ONE